# Genome-Wide Investigation of the NAC Transcription Factor Family in *Apocynum venetum* Revealed Their Synergistic Roles in Abiotic Stress Response and Trehalose Metabolism

**DOI:** 10.3390/ijms24054578

**Published:** 2023-02-26

**Authors:** Xiaoyu Huang, Xiaojun Qiu, Yue Wang, Aminu Shehu Abubakar, Ping Chen, Jikang Chen, Kunmei Chen, Chunming Yu, Xiaofei Wang, Gang Gao, Aiguo Zhu

**Affiliations:** 1Institute of Bast Fiber Crops, Chinese Academy of Agricultural Sciences, Changsha 410221, China; 2Department of Agronomy, Bayero University Kano, Kano PMB 3011, Nigeria; 3National Breeding Center for Bast Fiber Crops, Changsha 410221, China; 4Key Laboratory of Genetic Breeding and Microbial Processing for Bast Fiber Product of Hunan Province, Changsha 410221, China

**Keywords:** NAC transcription factor, drought stress, salt stress, trehalose metabolism pathway

## Abstract

NAC (NAM, ATAF1/2, and CUC2) transcription factors (TFs) are one of the most prominent plant-specific TF families and play essential roles in plant growth, development and adaptation to abiotic stress. Although the NAC gene family has been extensively characterized in many species, systematic analysis is still relatively lacking in *Apocynum venetum* (*A. venetum*). In this study, 74 AvNAC proteins were identified from the *A. venetum* genome and were classified into 16 subgroups. This classification was consistently supported by their gene structures, conserved motifs and subcellular localizations. Nucleotide substitution analysis (Ka/Ks) showed the *AvNACs* to be under the influence of strong purifying selection, and segmental duplication events were found to play the dominant roles in the AvNAC TF family expansion. *Cis*-elements analysis demonstrated that the light-, stress-, and phytohormone-responsive elements being dominant in the AvNAC promoters, and potential TFs including Dof, BBR-BPC, ERF and MIKC_MADS were visualized in the TF regulatory network. Among these AvNACs, AvNAC58 and AvNAC69 exhibited significant differential expression in response to drought and salt stresses. The protein interaction prediction further confirmed their potential roles in the trehalose metabolism pathway with respect to drought and salt resistance. This study provides a reference for further understanding the functional characteristics of *NAC* genes in the stress-response mechanism and development of *A. venetum*.

## 1. Introduction

In nature, plants are challenged by a variety of adverse abiotic stress conditions such as drought, salinity and extreme temperatures. These abiotic stresses limit the area of arable lands for agriculture and negatively affect crop productivity. To resist these stresses, plants have evolved complex and sophisticated regulatory pathways to sense and adapt to these stresses in a timely manner, which are regulated by promoting the expression of stress-responsive genes [1]. Transcription factors are the core of regulating gene expression by specifically binding to *cis*-elements of the target gene promoter to regulate the expression of downstream genes [2]. The NAC family first identified in *Petunia hybrida* [3] is a plant-specific TF family with its domain comprises a DNA-binding domain at N-terminal, a nuclear localization signal (NLS) and a transcriptional activation domain (AD) at C-terminal. The N-terminal region is a conserved domain containing about 160 amino acids. The C-terminal region, however, is highly variable, interacting with other transcription factors that may play a role in various developmental functions [4].

NAC TFs have been found to be involved in various growth and developmental processes, including cell division, seed development, root construction and control senescence [5,6,7,8,9,10,11]. In addition, they also positively response to salinity, drought, heat, cold and so forth [6,12,13,14]. The overexpression of *OsNAC5*, *OsNAC6*, *OsNAC9* and *OsNAC10* increased the root number and altered root architecture, thereby improving drought tolerance and grain yield in transgenic plants [15,16]. Rice NAC genes *ONAC022*, *ONAC045* and *ONAC066* were induced by drought, salt and abscisic acid (ABA) treatments, and positively regulated ABA-mediated pathway [17,18]. Similarly, overexpression of *LpNAC17* (from *Lilium pumilum*) in tobacco, *HhNAC54* (from *Hibiscus hamabo* Sieb. et Zucc.) in *Arabidopsis* enhanced salt tolerance, and *SlNAC10* (from *Suaeda liaotungensis*) in *Arabidopsis* in addition to salt enhanced drought tolerance [1,19,20]. Additionally, the NAC transcription factor *ATAF1* (Arabidopsis Transcription Activation Factor 1) was found to respond to carbon starvation by participating in trehalose metabolism. Overexpression of *ATAF1* directly activates the only trehalase-encoding gene *TREHALASE1*, reducing the trehalose-6-phosphate (Tre6P) levels and sugar starvation metabolome [21]. In contrast, rice NAC transcription factor OsNAC23 regulated carbon allocation by directly repressing the transcription of the trehalose-6-phosphate phosphatase (TPP) gene *TPP1* to increase Tre6P level and reduce trehalose content. Moreover, overexpression of *OsNAC23* gene in rice increased photosynthetic rate, sugar transportation and sink organ size, thus increasing grain yield [22]. These reports indicate that the NAC TFs are involved in forming a complicated regulatory network for plant response to external adversity with key roles in plant growth and development.

*Apocynum venetum* L., a member of the Apocynaceae family, is an important source of natural bast fiber. It has a wide range of pharmacological activities and is used for the prevention and treatment of cardiovascular and neurological diseases [23,24,25]. *A. venetum* is widely distributed throughout the saline–alkaline soils and sandy soils of northwestern China and the Mediterranean area [26]. It has a relatively high tolerance to various abiotic stresses, including drought, salt, cold, high temperature and wind, playing a crucial role in sand fixation and soil and water conservation [25]. Therefore, the exploration of the contributions of *A. venetum NACs* to drought and salinity tolerance is of great importance. However, a systematic analysis of NAC gene family in *A. venetum* has not been completed.

In the present study, genome-wide identification and analysis of the NAC family members were performed in the *A. venetum* genome. The physicochemical characteristics, chromosome localization, gene duplication, collinearity, phylogenetic relationship, cis-acting elements, gene structure and conserved motifs were analyzed comprehensively. Moreover, the protein–protein interaction prediction was also conducted to unravel their potential functions. The *NAC* gene expression pattern in various tissues were analyzed based on existing transcriptome data and, in response to drought and salt stresses, were investigated by quantitative real-time PCR (qPCR). Our results will contribute to further functional studies of the *NACs*, as well as provide a valuable reference for the genetic improvement of rooibos.

## 2. Results

### 2.1. Identification of NAC Family Members in A. venetum

A total of 74 NAC TF members named *AvNAC1–AvNAC74* according to their positions on the chromosomes were identified in *A. venetum* by the HMM search and complete domain analysis (Figure 1). *AvNAC* genes were unevenly distributed on chromosomes, with a high enrichment on chromosome 8 with 14 *AvNAC* members. Each of the *A. venetum* eleven chromosomes has at least three *AvNAC* members, reflecting its diversity and complexity, except for chromosome 5, which contained no *AvNAC* gene. The physicochemical properties of *AvNAC* genes are listed in Appendix A. The sequence length ranged from 164 to 688 amino acids with molecular weight from 19.36 to 76.26 kDa, and isoelectric point (pI) 4.39 to 9.58. Subcellular location prediction showed that sixty-four AvNAC proteins were nucleoprotein, eight members were cytoplasmic protein, and only one each in the chloroplast and cell membrane. These results suggested that AvNAC TFs might be involved in regulating the nuclear gene expression and have various functions to adapt to different environments.

Replication events, including tandem and segmental replication, are responsible for the expansion of gene families and the complexity of genomes during plant evolution. A duplication analysis of 74 *AvNAC* genes revealed four pairs of tandem duplicated genes distributed on three chromosomes (LG01, LG02 and LG08) (Figure 1), seven pairs of segmental duplicated genes unevenly distributed on the remaining chromosomes, except for LG05 and LG09 (Appendix A and Figure 2). In addition, the causes of divergence were measured by Ka and Ks, and Ka/Ks ratios analysis to determine the positive pressure after duplication (Appendix A). Normally, the Ka/Ks > 1 indicates positive selection, Ka/Ks = 1 represents neutral selection, while Ka/Ks ≤ 1 means purifying selection. The result showed that the Ka/Ks values of all segmental and tandem duplicated *AvNAC* gene pairs were less than 1, except for four gene pairs with significant sequence divergence and long evolutionary distance. These results indicated that most *AvNACs* evolved mainly under purifying selection, and the segmental duplications were the main driving force during the evolution process.

To further understand the evolutionary relationship between *A. venetum* and other plant species, we selected three dicotyledons (*Arabidopsis thaliana*, *Medicago sativa* and *Solanum lycopersicum*) and three monocotyledons (*Oryza sativa*, *Zea mays* and *Mauremys sinensis*) to establish collinearity analysis (Figure 3). The results showed *A. venetum* to have 63, 56 and 68 orthologous gene pairs with *A. thaliana*, *M. sativa* and *S. lycopersicum*, respectively. In contrast, there were fewer orthologous gene pairs between *A. venetum* and the monocotyledons *O. sativa*, *Z. mays* and *M. sinensis*, with 25, 18, and 22 pairs, respectively (Appendix A). Among these plant species, *A. venetum* had most orthologous gene pairs with *S. lycopersicum*, showing a closer evolutionary relationship, suggesting that the two species might have shared the similar phylogenetic divergence time. We found five orthologous gene pairs common to all the six species, suggesting that they might have existed before ancestral divergence. In addition, a total of 28 *AvNAC* genes were present only in dicotyledons, suggesting that these genes may have evolved after the divergence of the two classes of plants.

### 2.2. Phylogenetic Analysis of AvNACs

To explore the evolutionary relationship among the NAC TFs, we constructed a phylogenetic tree consisting of 179 NAC protein sequences (105 from *Arabidopsis* and 74 from *A. venetum*). A total of 74 AvNACs were divided into 16 subgroups according to the classification of NAC gene family in Arabidopsis (Figure 4). All groups contained AvNAC members except ANAC001, which only consisted of NACs from Arabidopsis (ANACs). ANAC063 contained the largest AvNAC members (20), while TIP, OsNAC8, SENU5 and AtNAC3 each had only one AvNAC member. Remarkably, the AvNAC members clustered into the same groups as ANACs, indicating higher homologies and may possibly have similar function. For example, some groups contained several ANACs that were known to be associated with stress responses, including ANAC19, ANAC55 and ANAC72 in group AtNAC3, ANAC56 in group NAP, and ANAC2 and ANAC81 in group ATAF. A total of eight AvNACs were distributed in these three groups, suggesting that they may respond to stress in *A. venetum*.

### 2.3. Gene Structure and Motif Analysis of AvNACs

To further investigate the relationship among 74 *AvNAC* genes, the phylogeny, gene structure and conserved motifs were analyzed (Figure 5). Fifteen conserved motifs were identified with amino acid lengths ranging from 11 to 99, and the number of motifs in the AvNAC proteins ranged from a minimum of three to a maximum of nine (Figure 5B). The most remarkable motifs were motif 1-6, which were found in 44 AvNAC proteins. Among all AvNAC proteins, AvNAC50, AvNAC53 and AvNAC54 contained nine motifs, whereas AvNAC46, AvNAC63, and AvNAC70 contained only three motifs. As expected, closely related members within the same subgroup possessed a relatively consistent motif composition, implying that they had similar functions. These conserved motifs may indicate potential functional site for the genes and, thus, may be involved in inducing similar downstream functions (Figure 5A). Motif 6 may be the most critical motif since it was present in all proteins but AvNAC46. In contrast, motif 13 was the least common and was only found in AvNAC11. The variable distribution of motifs may contribute to the functional diversity of the AvNAC gene family.

A gene structure analysis revealed that the intron number of AvNAC genes ranged from 0 to 10, of which seven genes lacked intron, while *AvNAC67* contained 10 introns (Figure 5C). The number and distribution of exon-intron within the same subgroup displayed some similarities; for example, *AvNAC29* and *AvNAC20* are assembled together in the phylogeny tree, both of them had two introns with similar distribution patterns. Notably, most of the intron insertions occurred in the conserved domains of the NAM, indicating their importance in plants. These results not only strongly support the reliability of the classification but also further reveal that *AvNACs* within the same group may play similar functional roles.

### 2.4. Cis-Element Analysis of AvNACs

To explore the potential function of *AvNACs*, the prediction of *cis*-element 1500 bp upstream of the *AvNACs* transcription start site was performed. A total of 51 types of *cis*-elements were detected and classified into four categories, among which the most abundant was light responsive (19 types), followed by phytohormone responsive (13 types), stress-responsive (13 types) and plant growth and development (6 types) (Figure 6 and Appendix A). Among the light-responsive, Box 4 was the dominant element followed by G-box. The cis-elements responsive to phytohormone were mainly methyl jasmonate (MeJA), abscisic acid (ABA), ethylene (ERE) and gibberellic acid (GA) responsive elements. Among them, ABRE elements were the most abundant (104 in total), followed by ERE (101 in total), TGACG-motifs (61 in total, MeJA responsive) and CGTCA-motifs (61 in total, MeJA responsive). In terms of stress responsiveness, MYC elements (179 in total, drought responsiveness) were the prominent elements, appearing in 68 *AvNAC* genes, followed by STRE (106 in total, stress responsiveness). Notably, drought-responsive elements were present in almost all *AvNAC* genes except *AvNAC34* and *AvNAC43*. At the same time, other elements important for stress response, including cold/dehydration responsive, wound responsive, low-temperature and anaerobic induction elements, were also detected. In addition, several elements related to plant growth and development were identified, with seed-specific expression elements being the most abundant. Notably, the *AvNAC* genes with similar cis-elements types and numbers were shown to have close phylogenetic relationships, such as *AvNAC4* (11 ABRE and 12 G-box), *AvNAC58* (10 ABRE and 11 G-box) and *AvNAC69* (5 ABRE and 5 G-box), suggesting that this subgroup of *AvNAC* genes might be important in ABA signaling pathway and light response. These results further reveal the potential function of the *AvNAC* genes and their roles in plant development and environmental stress responses.

### 2.5. Expression Pattern of AvNAC Genes

To understand the expression patterns of the *AvNACs*, transcript levels in different tissues (root, stem and leaf) were analyzed based on the transcriptome data (SMAN23766539, SMAN23766540 and SMAN23766541). The results showed that thirty-five of the *AvNACs* were expressed in all the three tissues, with *AvNAC11*, *AvNAC15* and *AvNAC16* showing significant expression levels (Figure 7), indicating that these genes may widely participate and play an important role in plant growth and development. Fourteen of the *AvNACs* showed tissue-specific expression, five expressed in only one tissue and nine in two tissues. For example, *AvNAC31* had a higher expression level in stem and leaf, *AvNAC71* in root and stem, *AvNAC35* in root and leaf, whereas *AvNAC3*, *AvNAC28* and *AvNAC74* were only expressed in root. Twenty-five of the *AvNACs* were not expressed in any tissue, suggesting that the expression of these genes requires a specific development stage or environmental induction. Taken together, these findings indicate that different *AvNACs* play distinct roles in different tissues.

Based on phylogenetic analysis and homology with known *NAC* genes in *A. thaliana*, 15 and 20 *AvNAC* genes associated with drought and salt stress response were selected, respectively, and their expression patterns under drought and salt stress were investigated by qPCR. Under drought stress, *AvNAC1* had a similar expression pattern in the leaf, stem and root. Its expression improved with increasing PEG concentration (from 0 to 20%) and reached the highest level at 20%, where the expression level in stem and leaf was 200- and 100-fold higher than that of the control, respectively (Figure 8 and Appendix A). The expression of *AvNAC4* and *AvNAC43* was also induced by drought stress. *AvNAC4* in leaf and *AvNAC43* in root and stem increased, while *AvNAC4* in root and stem and *AvNAC43* in leaf showed a trend of increasing and then decreasing, but were still higher than the control. Conversely, the expression levels of some genes gradually decreased with increasing PEG concentration, including *AvNAC6*, *AvNAC10* and *AvNAC23* in the root, and *AvNAC2* in the leaf. *AvNAC58* and *AvNAC69* showed a similar expression pattern in root, decreasing rapidly and then increasing, but overall was lower than the control. In contrast, their expression in the stem and leaf was generally elevated. Notably, some genes were highly expressed in specific tissue. For example, under 5% PEG treatment, *AvNAC32* in root and *AvNAC43* in leaf were 150- and 100-fold higher than the control, while under 10% PEG treatment, *AvNAC6* in leaves and *AvNAC43* in root were 100- and 80-fold higher than the control, respectively.

Under salt stress, *AvNAC58* and *AvNAC69* showed similar expression patterns, with increasing NaCl concentrations (from 0 to 200 mM), their expression in all analyzed tissues were gradually increased, and reached the highest level at 200 mM (Figure 9 and Appendix A). The expression of *AvNAC4*, *AvNAC43* and *AvNAC44* in all analyzed tissues were also induced by salt stress. Among them, the expression of *AvNAC4* in root, *AvNAC43* in root and stem, and *AvNAC44* in stem and leaf all reached the highest at 100 mM NaCl treatment and then decreased, but were higher than the control. Otherwise, their expression in other tissues gradually increased and reached the highest at 200 mM NaCl treatment. In contrast, the expression of *AvNAC6* in root and stem decreased with increasing NaCl concentration, reaching a minimum at 200 mM treatment. Similar situations were observed in the expression of *AvNAC23* in the root and *AvNAC2* in the leaf. Notably, the expression of some *AvNAC* genes was significantly increased in root under salt stress, including *AvNAC1*, *AvNAC4*, *AvNAC7*, *AvNAC32* and *AvNAC69*. Among them, the expression levels of *AvNAC1* and *AvNAC69* were 100-fold higher than the control under 200 mM NaCl treatment, and the expression levels of *AvNAC4*, *AvNAC7* and *AvNAC32* were 200-, 100- and 40-fold higher than the control under 100 mM NaCl treatment, respectively.

Taken together, some genes showed similar expression patterns under drought and salt stress; for example, *AvNAC4* and *AvNAC43* showed increased expression levels in roots, stems and leaves, especially in roots. In contrast, the expression patterns of *AvNAC58* and *AvNAC69* in roots were different under different stresses, decreasing under drought stress while increasing under salt stress. Moreover, some genes, including *AvNAC32* and *AvNAC43*, were only expressed under stresses. These results suggest that *AvNAC* genes possess various expression patterns under different stresses, indicating their functional specificity. In addition, we found that the expression levels of many *AvNAC* genes were significantly altered in roots under stresses, including *AvNAC1*, *AvNAC4*, *AvNAC32, AvNAC43* and *AvNAC69*, indicating their important roles in stress resistance in *A. venetum*.

### 2.6. Protein–Protein Interaction and Regulatory Network Analysis Analysis of AvNAC

To further explore the functions of the *AvNAC* genes, a protein–protein interaction network of AvNAC protein was performed based on their homologs in *A. thaliana*. As shown in Figure 10, the majority of the AvNAC proteins are homologous and interact with known *Arabidopsis* proteins, including AtNST1, AtXND1, AtNAC073, AtNAC083, AtRD26, AtNAC1, AtVND1/7, AtCUC2/3, AtNAC007 and AtNAC059, of which different groups may had different functions. AvNAC4 was homologous to AtRD26, which interacted with KIN10 (SnRK1.1) and SOS2 protein, while AvNAC32 was homologous to AtNAC047, which interacted with CIPK20 (SOS3) protein. SOS2 and SOS3 are important protein kinases of SOS pathway, which control the expression and activity of SOS1 to regulate salt stress response in plant [27]. Both ATAF1 and ATAF2 interacted with KIN10 (SnRK1.1) and KIN11 (SnRK1.2), which in turn interacted with TPS1. In addition, ATAF1 also interacted with TPPA and TPPJ, which belongs to a cluster of proteins related to the trehalose metabolism pathway. The result indicates that the homologs of these proteins in *A. venetum* may also possess similar functions.

TFs can regulate gene expression by binding to specific sequences upstream of the start codon of target genes. Potential TFs were investigated in the upstream regions of all 74 *AvNAC* genes and a TF regulatory network was established. A total of 3820 TFs were detected and belonged to 40 TF families (Figure 11A and Appendix A). Among these TF families, Dof (637) contained the maximum number of members followed by BBR-BPC (593), ERF (497), NAC (312) and MIKC_MADS (298), while WOX (4), LFY (2) and RAV (2) contained only a few members. Among 74 *AvNAC* genes, *AvNAC58* was targeted by 206 TFs, which was the most abundant, and followed by *AvNAC9* (198), *AvNAC64* (196), *AvNAC23* (188) and *AvNAC19* (147) (Figure 11B and Appendix A). Furthermore, the *AvNAC* genes were targeted by different types and numbers of TF families that are associated with plant growth, development and response to biotic/abiotic stress. For instance, *AvNAC58* was targeted by ERF, BBR_BPC, bZIP, bHLH, TCP and MYB family simultaneously.

## 3. Discussion

### 3.1. Global Profile of NAC Gene Family in A. venetum

In the present study, 74 *NAC* genes were identified from *A. venetum*, divided into 16 subgroups and distributed unevenly on 11 chromosomes (Figure 1, Figure 2 and Figure 4). A majority of *AvNACs* (64/74) were predicted to be nuclear proteins that may respond to drought stress by regulating the expression of genes related to stress-, lipid transport-, and lipid localization-related genes (Appendix A). Gene structural diversity is significant for gene evolution. *AvNAC* genes within the same subgroup shared a similar intron/exon composition, and the proteins they encode had a similar motif component (Figure 5), which was consistent with what was reported in sunflower [7], *Vigna radiata* L. [28] and *Zanthoxylum bungeanum* [29], suggesting that the NAC gene family was highly conserved. In particular, among the 15 motifs identified, motif 13 appeared in three NAC members clustered in the same subgroup and it was predicted to be localized in the cytoplasm. Similar situations were found in motif 15, suggesting that these members may possess a specific function.

The number of AvNAC gene family member was the same as in *Vitis vinifera* (74) [30], more than in *Lagerstroemia indica* (21) [31] and *Lolium perenne* (72) [1] but less than in *O. sativa* (151) [8], *Z. mays* (152) [9], *Vigna radiata* L. (81) [28] and *Passiflora edulis* (105) [32]. The variation may be related to gene duplication during the evolution of the species. Segmental and tandem duplication are the dominant driving forces of evolution and expansion of family genes [33]. We identified seven segmental pairs and four tandem pairs in *A. venetum* (Figure 1 and Figure 2). A total of 15 segmental duplications were reported in the mung bean (*V. radiata*) NAC gene family [28]. *ZbNAC* (*Z. bungeanum*) gene family had forty-two segmental duplication pairs and nine tandem duplication pairs [29]. A total of one-hundred-and-twenty-one pairs of segmental duplication and nine pairs of tandem duplication pairs were presented in the *M. sinensis* NAC gene family [34]. Therefore, it is speculated that the segmental duplication is the dominant force driving the evolution and expansion of NAC gene family. Selective pressure analysis revealed that *AvNAC* genes evolved under purifying selection (Appendix A). Additionally, the collinearity analysis (Figure 3) indicated that extensive evolution and duplication of the *AvNAC* genes may have occurred after the divergence of monocotyledons and dicotyledons.

### 3.2. Roles of AvNACs in Drought and Salt Stress

Abiotic stress triggers a wide range of plant responses, including the expression of related genes, accumulation of metabolites, plant growth and development state, and crop yield changes. Thus, mining key genes for drought and salt resistance is of great importance to agricultural production and provides theoretical data for further research on the mechanism. Considering the strong drought and salinity tolerance in *A. venetum*, we treated the species with varying drought and salt concentrations to study the expression pattern of *AvNAC* genes under these stresses, respectively. The expression of *AvNAC4* in all analyzed tissues increased under both drought and salt stresses (Figure 8 and Figure 9). It was classified into the ATNAC3 subgroup along with *ANAC019*, *ANAC055* (*AtNAC3*) and *ANAC072* (*RD26*), which were reported to be induced by drought, high salinity and ABA, and their overexpression improved the drought tolerance of transgenic plants [35]. Moreover, *AvNAC4* was homologous to *ANAC072* (*RD26*) (Figure 10), which is thought to be involved in a novel ABA-dependent stress signal pathway [36]. Under drought stress, ABA primarily promotes stomatal closure to minimize transpiration, while activating stress-responsive genes that work together to improve plant stress tolerance [37]. In addition, more studies reported that ABA enables plants to recover from salt stress and water-deficit environment by increasing hydraulic conductivity or promoting root cell elongation [37,38,39]. This might be attributed to the fact that the expression levels of *AvNAC4* in roots significantly increased under salt and drought stress, suggesting that *AvNAC4* might be involved in stress responses and ABA signal pathways to enhance plant tolerance. The substantial presence of ABRE and MYC elements in the *AvNAC4* promoter region (Figure 6) further supported its involvement in stress responses.

In addition, the expression levels of *AvNAC58* and *AvNAC69* were increased in leaf and stem but decreased in root under drought stress, while they were increased in all analyzed tissues under salt stress (Figure 8 and Figure 9). *AvNAC58* and *AvANC69,* clustered in the ATAF subgroup, were homologous to *ANAC002* (*ATAF1*) and *ANAC081* (*ATAF2*), respectively (Figure 4 and Figure 10). The overexpression of *ATAF1*, *ATAF2* or their homologues in plants were reported to improve the transcription of some stress-related genes, thus enhancing the tolerance of plants to drought or salt stress [40,41]. Additionally, allogeneic overexpression of *CaNAC46* (*Capsicum annuum*), belonging to the ATAF family in *Arabidopsis*, increased the tolerance of transgenic plants to drought and salt stress as well [42]. Thus, *AvNAC58* and *AvNAC69* may also have the same function. Our research found that there were multiple TFBSs in the promoter region of *AvNAC* genes, among which *AvNAC58* was simultaneously targeted by Dof, ERF, BBR_BPC, bZIP, bHLH, TCP and MYB family for a total of 206 TFs, which was the most abundant (Figure 11 and Appendix A). Dof, AP2/ERF, bZIP, MYB and bHLH TFs play important regulatory roles in various abiotic stresses [43,44,45,46,47]. These results supported that *AvNAC58* and *AvNAC69* are involved in drought and salt stress responses, and they may have different mechanisms of action under various stresses.

### 3.3. AvNACs Regulate Plant Drought and Salt Tolerance through Trehalose Pathway

ATAF1 interacts with the catalytic subunits AKIN10 and AKIN11 of SnRK1 (SNF1-RELATED KINASE 1), and it is a key regulator of ABA signaling pathways [48]. Our protein–protein interaction analysis further confirmed this, where ATAF1 (AvNAC58) and ATAF2 (AvNAC69) interacted with AKIN10 and AKIN11. In addition, RD26 was predicted to interact with AKIN10 (Figure 10). Notably, AKIN10 and AKIN11 interact with ATTPS1, ATAF1 interacts with ATTPPA and ATTPPD, while *TPS* and *TPP* are essential enzyme genes in trehalose biosynthesis pathway in plants [49]. This further associates the stress response with trehalose metabolism. Moreover, *ATAF1* and *ANAC032* regulate trehalose metabolism through the direct regulation of TRE1 expression, while *OsNAC23* regulates Tre6P and trehalose levels by repressing TPP1 transcription, all of which are induced by carbon starvation [21,22,41]. These results indicated that *AvNAC4*, *AvNAC58* and *AvNAC69* might enhance the stress tolerance of *A. venetum* by participating in the trehalose metabolism.

In plants, the trehalose biosynthetic pathway plays a key role in regulating carbon allocation and stress adaptation [50]. Meanwhile, trehalose is an important signal linking plant metabolism, growth and development [49]. Tre6P, an intermediate product of trehalose synthesis, is involved in photosynthesis regulation, embryo formation, cell differentiation and starch synthesis. The upregulation of *TRE1* expression leads to a decrease in Tre6P and trehalose level, which results in the activation of SnRK1 activity that is inhibited by Tre6P [51]. TPS is a key enzyme gene for Tre6P synthesis and has a central role in trehalose biosynthesis [49]. Taken together, SnRK1 interacts with ATAF1, ATAF2, and TPS1, which links stress response, ABA signaling pathway and trehalose metabolism. On the one hand, *AvNAC4, AvNAC58* and *AvNAC69* may be involved in the ABA signaling pathway through interaction with the key regulator SnRK1. On the other hand, *AvNAC58* and *AvNAC69* may be involved in trehalose metabolism pathway through direct regulation of *TRE1*, while the decreased Tre6P level alleviated its inhibition of SnRK1 activity, thus affecting the ABA signaling pathway.

In summary, plant resistance to environmental stress is an integrated regulatory network. The ability to withstand drought and salt stress is also the result of the combined action of various mechanisms. *AvNAC4, AvNAC58* and *AvNAC69* are potential candidate genes for improving the drought and salt tolerance of *A. venetum* and further studies are needed for verification.

## 4. Materials and Methods

### 4.1. Identification and Bioinformatic Analysis of NAC Gene Family in A. venetum

The NAC TF members were identified from the *A. venetum* genome derived from the whole genome data sequenced by our laboratory. To find the putative members of the NAC family in *A. venetum*, we used two methods, HMM (Hidden Markov Models) and BLASTp (Basic Local Alignment Search Tool for proteins). The HMM file of the NAM domain (PF02365) was obtained from Pfam (https://pfam.xfam.org/, accessed on 19 September 2022) [52], and the target genes were searched using HMMER 3.0 software with a threshold of e-value ≤ 10^−5^. Meanwhile, the NAC protein sequences of Arabidopsis (ANACs) were downloaded from TAIR (The Arabidopsis Information Resource, https://www.arabidopsis.org/, accessed on 19 September 2022) [53] and blast against the *A. venetum* genome by local BLASTp program with e-value ≤ 10^−5^. The outcomes of the two methods were merged and further verified by NCBI batch CD-search (https://www.ncbi.nlm.nih.gov/cdd (accessed on 20 September 2022)) with e-value ≤ 10^−5^ and SMART (Simple Modular Architecture Research Tool, http://smart.embl-heidelberg.de/, accessed on 29 September 2022) [54] to ensure that the sequences contained the complete NAM domain.

The physical and chemical properties of the AvNAC proteins including amino acid number, molecular weight (MW) and isoelectric point (pI), were analyzed by ExPasy (https://www.expasy.org/, accessed on 11 October 2022) [55]. The CELLO v.2.5 (http://cello.life.nctu.edu.tw/, accessed on 11 October 2022) [56] was utilized for subcellular localization prediction of the AvNACs.

The amino acid sequences of ANACs and identified AvNACs were merged and then aligned using MAFFT (L-INS-i algorithm) [57]. Subsequently, a phylogeny was generated by MEGA 11 with the Poisson model, using the neighbor-joining method with 1000 bootstraps and pairwise gaps deletion [58]. According to the classification of ANACs, the *A. venetum* NAC proteins were sub-divided into different subgroups. The iTOL (Interactive Tree of Life, https://itol.embl.de/, accessed on 20 October 2022) [59] was used to visualize the phylogenetic tree.

Gene structures of *AvNACs* were characterized by the GSDS (Gene Structure Display Server, http://gsds.gao-lab.org/, accessed on 27 October 2022) [60] and the conserved motifs of AvNACs were identified using MEME (Multiple Em for Motif Elicitation, https://meme-suite.org/meme/tools/meme, accessed on 28 October 2022) [61] with the following parameters: maximal e-value = 1 × 10^−5^ and a range of motif widths from 6 to 50.

The 1500 bp upstream sequences of the start codon (ATG) of each *AvNAC* gene were extracted from the *A. venetum* genome data using TBtools [62] and subjected to *cis*-element analysis using PlantCARE (https://bioinformatics.psb.ugent.be/webtools/plantcare/html/, accessed on 03 November 2022) [63]. Additionally, the 1,000 bp upstream sequences of each *AvNAC* gene were subjected to transcription factor binding sites prediction analysis using PTRM (Plant Transcriptional Regulatory Map, http://plantregmap.gao-lab.org/binding_site_prediction.php, accessed on 08 November 2022) online tool with p-value ≤ 1 × 10^−5^, and the regulatory network was constructed by Cytoscape v.3.9 software [64].

Prediction of protein–protein interaction was performed by the online program STRING v.11.5 (https://cn.string-db.org/, accessed on 23 November 2022) [65], using AvNAC proteins as the queries and the Arabidopsis proteins as references.

### 4.2. Chromosomal Locations, Gene Duplications, and Selection Pressure Analysis

The chromosome distribution information was acquired from the *A. venetum* genome annotations, and the chromosomal map with gene positions was drafted using TBtools [62]. All the *A. venetum* protein sequences were aligned using the local BLASTp program with e-value ≤ 1 × 10^−5^, number of alignments of 5. Then, the BLASTp result and GFF file of genomic annotation were used to generate collinearity and tandem files by using MCScanX software to screen segmental and tandem duplication genes of *AvNACs* [66]. The non-synonymous (Ka) and synonymous (Ks) values of duplicated *AvNAC* gene pairs were calculated using KaKs Calculator 2.0 software and the selection model of gene pairs was estimated based on the Ka/Ks ratio [67]. The synteny analysis map between *A. venetum* and other species was constructed using TBtools [62].

### 4.3. Expression Analysis of AvNAC Genes

The transcriptome data of three tissues comprising root, stem and leaf of *A. venetum* were obtained (SMAN23766539, SMAN23766540 and SMAN23766541) and the FPKM (fragments per kilobase of transcript per million mapped reads) values representing the expression levels of *AvNACs* were extracted to generate the heatmap by TBtools [62].

### 4.4. Plant Materials and Stress Treatment

*A. venetum* seedlings were cultured under a hydroponic system following the procedure [68]. Three-week-old seedlings of similar size were selected and divided into two groups, one for drought treatment and the other for salt treatment. The seedlings were subjected to drought- (5, 10, and 20% PEG6000) and salt- (50, 100, and 200 mM NaCl) stress treatment for 24 h, while the control was treated with only water. The roots, stems and leaves were collected, frozen in liquid nitrogen and stored at −80 °C.

### 4.5. RNA Isolation and qPCR

Total RNA was extracted using SteadyPure Plant RNA Extraction Kit (Accurate Biotechnology (Changsha, China) Co., Ltd.), and the purity and concentration of total RNA were measured with NanoDrop 2000 spectrophotometer (Thermo Scientific, Wilmington, DE, USA). First-strand cDNA was synthesized using Evo M-MLV One Step RT-PCR Kit and used as a template for qPCR together with gene-specific primers. The b-tubulin (Tu) gene was used as a reference [69] (Appendix A). Based on the reported NAC members associated with drought and salt stress in Arabidopsis and rice, the AvNAC genes were selected for qPCR by searching for their homologous sequences in *A. venetum*. qPCR was performed on CFX96 Touch Deep Well Real-Time PCR System (Bio-Rad, USA) using SYBR^®^ Green Premix Pro Taq HS qPCR Kit II (Accurate Biotechnology (Changsha, China) Co., Ltd.). The reaction procedure consisted of an initial denaturation at 95 °C for 30 s, followed by 40 cycles for 5 s at 95 °C and 60 °C for 30 s. All experiments were performed in three independent biological replicates and each in three technical replicates. The relative gene expression level was calculated by the 2^−ΔΔCT^ method.

## 5. Conclusions

A genome-wide identification and analysis of the NAC family members were performed in the *A. venetum* genome. In this research, a total of 74 AvNAC TF members were identified and classified into 16 subgroups with NAC proteins from *A. thaliana*, and this classification was consistently supported by further analysis, including gene structures, conserved motifs and subcellular localizations. The inter-species synteny analysis of *NAC* genes indicated the close evolutionary relationship between *A. venetum* and *S. lycopersicum*. Analysis of gene promoter *cis*-elements revealed the potential function of the *AvNAC* genes and their roles in plant development and environmental stress responses. Moreover, ten *AvNACs* exhibited significant differential expression in response to drought and salt stresses, and the protein interaction prediction further confirmed the potential roles of AvNAC58 and AvNAC69 proteins in the trehalose metabolism pathway with respect to drought and salt resistance. In general, these results provide a reference for further functional studies of the *NAC* genes and promote the genetic improvement of growth, development and stress resistance in *A. venetum*.

## Figures and Tables

**Figure 1 ijms-24-04578-f001:**
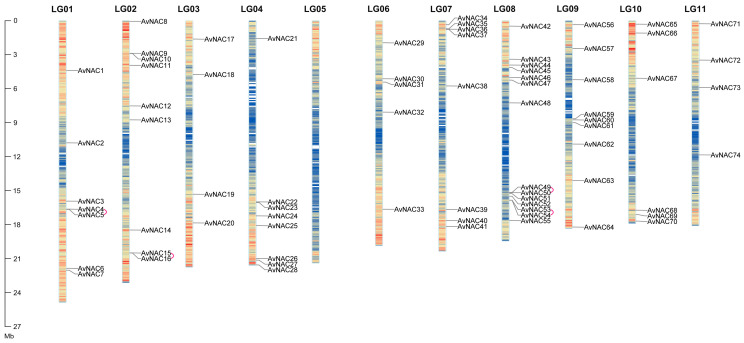
The distribution of *AvNACs* on chromosomes. Heatmaps on chromosomes represent gene density, from low to high in blue and red, respectively. The gene pairs of tandem duplication are marked with pink connecting lines.

**Figure 2 ijms-24-04578-f002:**
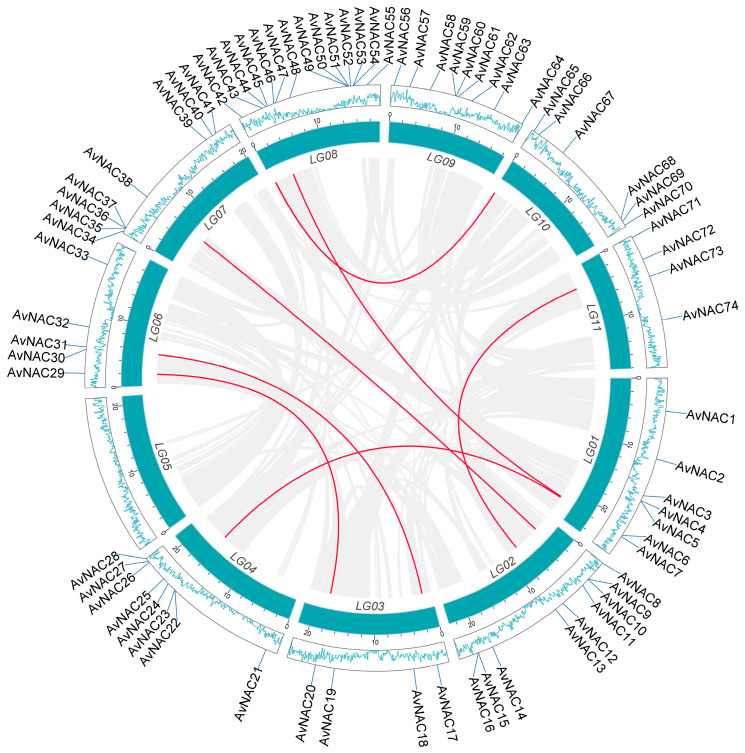
Collinearity analysis of the *AvNAC* genes. The blue box represents the chromosome, the lines inside the circle indicate collinear blocks within the *A. venetum* chromosome. The red lines indicate segment duplication events related to *AvNAC* genes. The blue lines inside the outer blocks indicate gene density.

**Figure 3 ijms-24-04578-f003:**
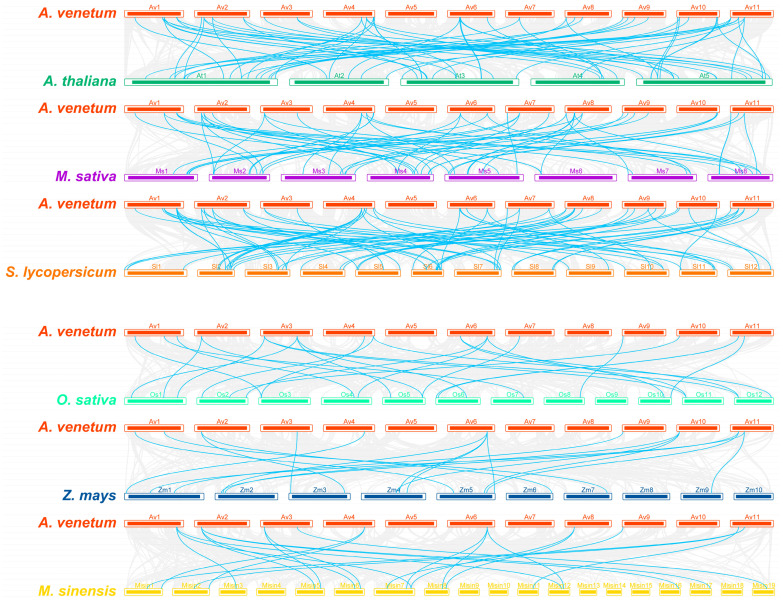
Synteny analysis of *NAC* genes between *A. venetum* and six other plant species (*A. thaliana*, *M. sativa*, *S. lycopersicum*, *O. sativa*, *Z. mays* and *M. sinensis*). The gray lines in the background represent collinear relationships between *A. venetum* and six other species, while the cyan lines highlight the syntenic *NAC* gene pairs.

**Figure 4 ijms-24-04578-f004:**
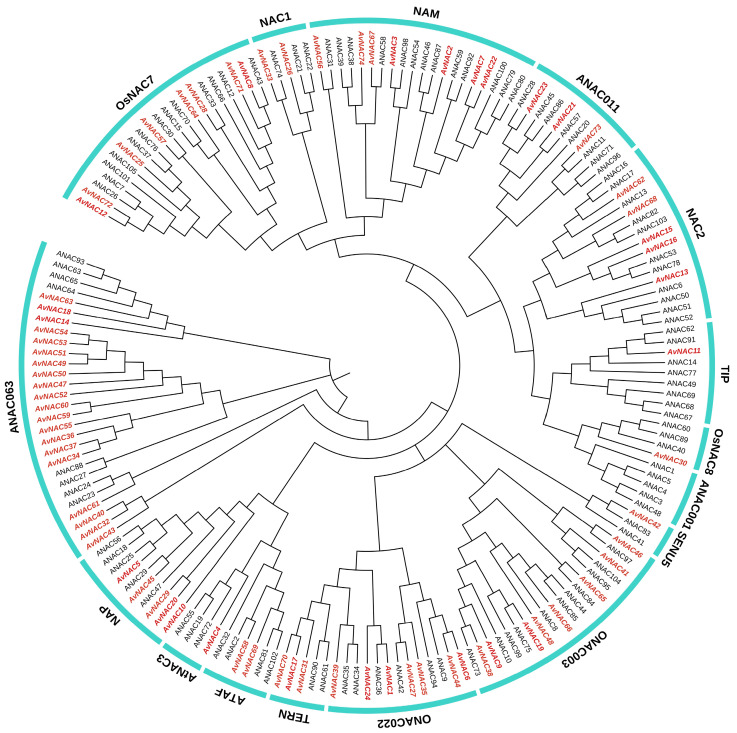
The phylogenetic tree of AvNAC proteins with AtNAC proteins in *A. thaliana*. The AvNACs are indicated by red color and italic, and the AtNACs are indicated by black color.

**Figure 5 ijms-24-04578-f005:**
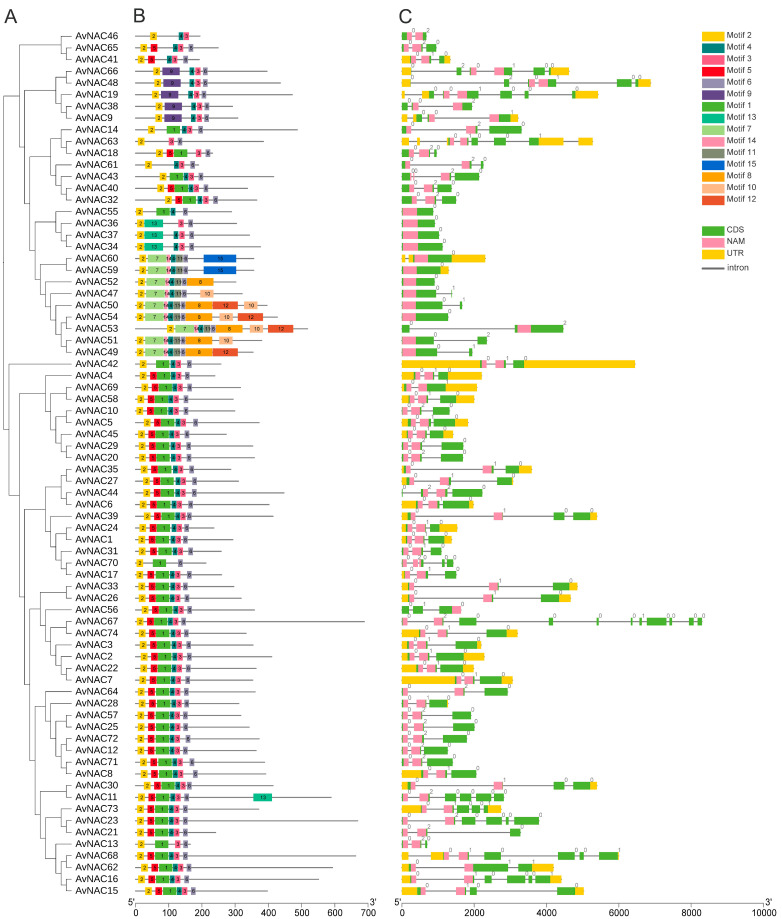
Phylogenetic tree, conserved motifs and gene structures of 74 AvNAC TFs. (**A**) Phylogenetic tree of 74 AvNAC TFs. (**B**) Conserved motifs in the 74 AvNAC proteins. Different colors represent different motifs. (**C**) Gene structures of *AvNAC* genes. Green and yellow boxes indicate the exons and untranslated regions (UTRs), and black lines indicate introns. Pink boxes highlight the NAM domains.

**Figure 6 ijms-24-04578-f006:**
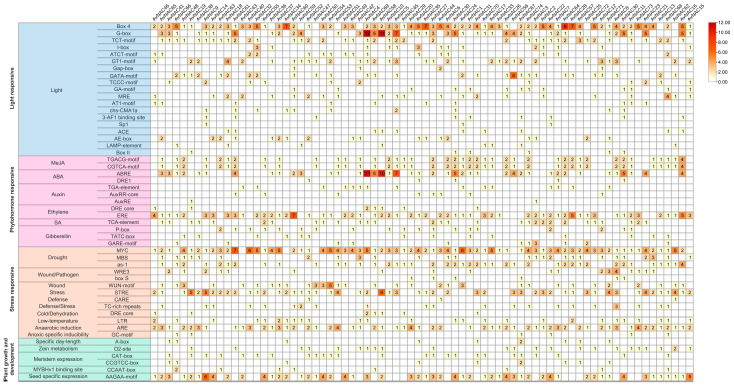
Analysis of *cis*-elements in AvNAC promoters. Statistics and categories of *cis*-elements in the promoter regions of *AvNAC* genes. The numbers of *cis*-acting element were indicated by numbers and a color gradient.

**Figure 7 ijms-24-04578-f007:**
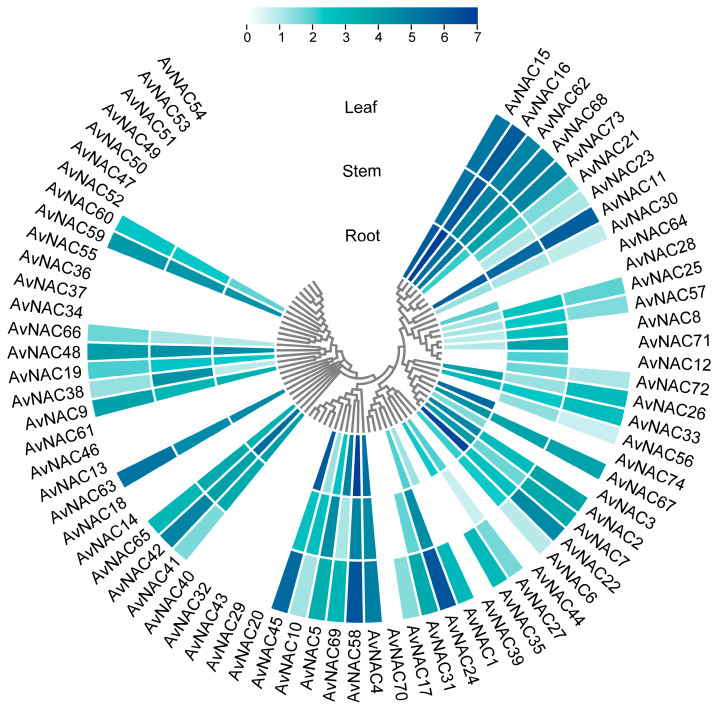
Expression patterns of *AvNAC* genes in different tissues (root, stem and leaf). Log_2_ (FPKM + 1) values are indicated by the shade of the color, with darker colors representing larger values.

**Figure 8 ijms-24-04578-f008:**
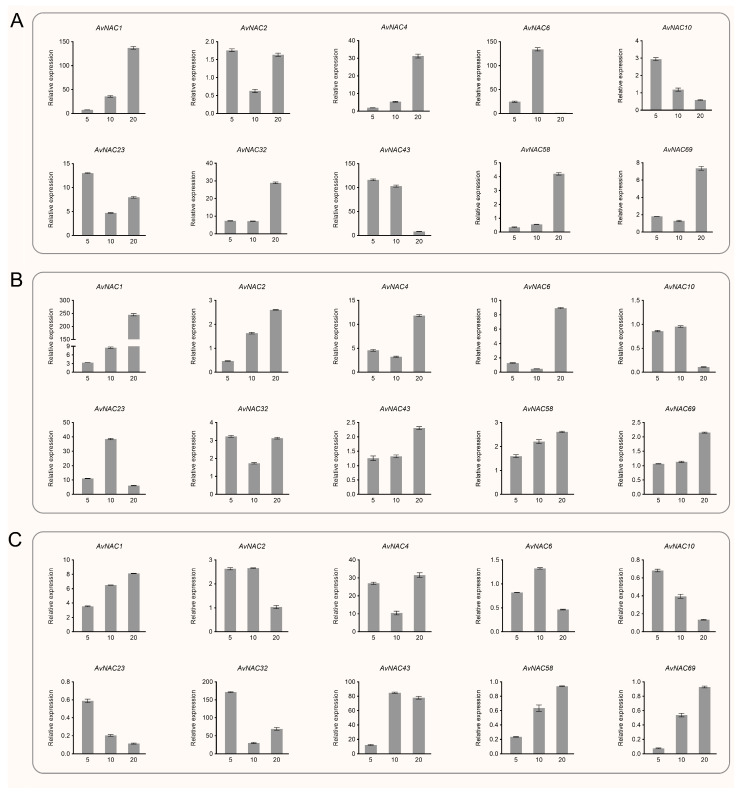
Relative expression of selected *AvNAC* genes in various tissues of *A. venetum* under drought stress (0, 5, 10, 20% PEG6000 treatment). (**A**) Leaf. (**B**) Stem. (**C**) Root.

**Figure 9 ijms-24-04578-f009:**
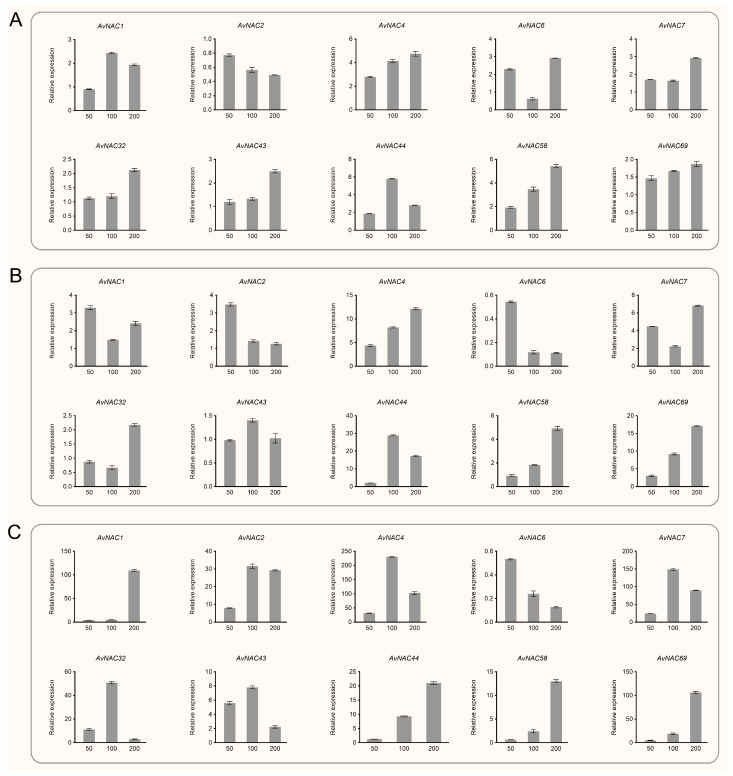
Relative expression of selected *AvNAC* genes in various tissues of *A. venetum* under salt stress (0, 50, 100, 200 mM NaCl treatment). (**A**) Leaf. (**B**) Stem. (**C**) Root.

**Figure 10 ijms-24-04578-f010:**
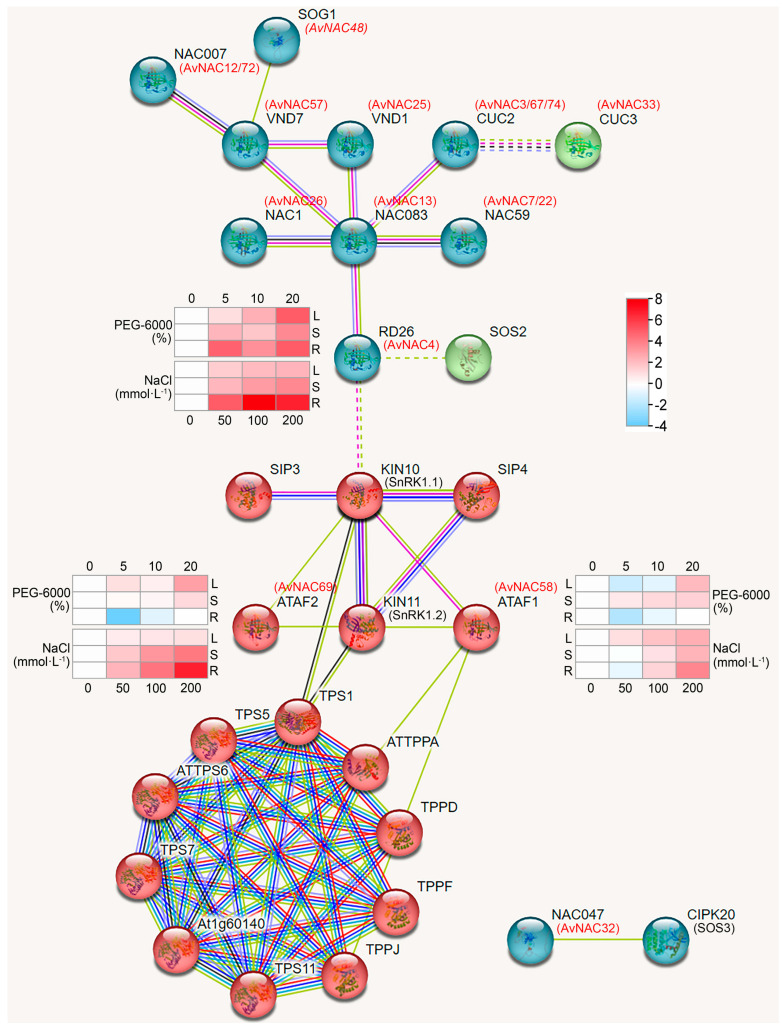
Protein-protein interaction of AvNAC proteins.

**Figure 11 ijms-24-04578-f011:**
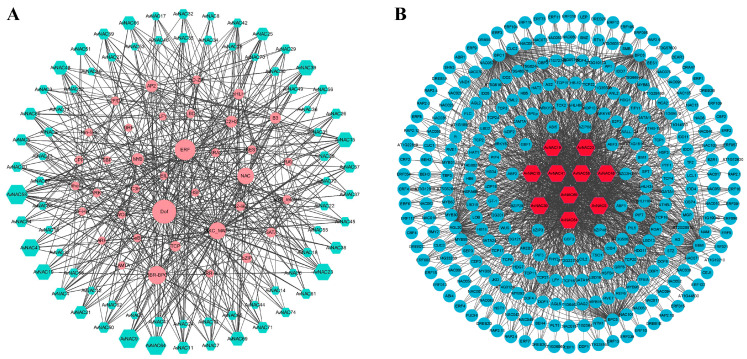
The putative transcription factor regulatory network analysis of *AvNAC* genes. (**A**) Interaction network of NAC genes and putative transcription factors in *A. venetum*. The node size represents the number of interactions. (**B**) The top 10 highly enriched and targeted *AvNAC* genes.

## Data Availability

The datasets used and/or analyzed during the current study are available from the corresponding author on reasonable request.

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
