# Peer review of "Genome-Wide Investigation of the NAC Transcription Factor Family in Apocynum venetum Revealed Their Synergistic Roles in Abiotic Stress Response and Trehalose Metabolism"

_ijms, 2023, doi:10.3390/ijms24054578_

Round 1

Reviewer 1 Report

Extensive, positive, important and well documented research, in my opinion. Especially pleased that you included two different stressors in your work with your target species--and that you compared it to three dicots and three monocots. Well conceived and well written--appropriate inferences drawn.

My most important question was about your acronym-rich abstract. Not sure if other reviewers will agree with my wish that your abstract spell out (or generalize) some of these concepts.

A number of comments and edits included in the attached manuscript copy.

Reviewer 2 Report

To,

The Editor,

IJMS, MDPI,

Manuscript ID: ijms-2228973

Subject: Submission of comments of the manuscript in “IJMS"

Dear Editor IJMS, MDPI,

Thank you very much for the invitation to consider a potential reviewer for the manuscript (ID: ijms-2228973). My comments responses are furnished below as per each reviewer’s comments. 

In the reviewed manuscript, the authors identified 74 AvNAC proteins were identified from the A. venetum genome and were classified into 16 subgroups. Segmental duplication events were found to play the dominant roles in the AvNAC TF family expansion. Nucleotide substitution analysis (Ka/Ks) showed the AvNACs to be under the influence of strong purifying selection. Fifty-one types of cis-elements were identified in the AvNAC promoters, with light-, stress-, and phytohormone-responsive elements being dominant. Potential TFs including Dof, BBR-BPC, ERF and MIKC_MADS were identified in the promoter region of all 74 AvNAC genes and visualized in a TF regulatory network. Ten AvNACs exhibited significant differential expression in response to drought and salt stresses. The protein interaction prediction further confirmed the potential roles of AvNAC58 and AvNAC69 proteins in the trehalose metabolism pathway with respect to drought and salt resistance. This study provides a reference for further understanding the functional characteristics of NAC genes in stress-response mechanism and development of A.venetum. The manuscript provide the important information on AvNAC gene family. Therefore, it might be conditionally accepted subject to major revision. Authors need to address the following issues before it can be accepted for publication.

  1. I have read the entire manuscript and my initial comment is that manuscript is poorly written. I have significant concerns about the grammar and vocabulary of the manuscript; therefore, I recommend the authors to used an English proofreading service.
  2. The structure of the abstract should be improved, as well as the lack of several aspects that should be included in this section. Most of the abstracts contain confusing and uninformative sentences. Please give more precise objectives here (such as in the Abstract). The abstract should highlight the most important results of the parameters and characteristics assayed.
  3. Introduction grammatical issues appear to be most prevalent in the introduction, making for very confusing reading. Further, the introduction is short but has no clear thread.
  4. General note: the figures in this section are quite low resolution and difficult to make out. Higher-resolution versions will be needed for publication, for example, in Figures 1, 2, 3, 4, 5, 6, 8, 9, and 11. Further, figure texts are not readable.
  5. In Material and Methods:- indicate how many replicates assayed in each analysis/parameter. The number of samples or biological and technical replicates should be mentioned for each parameter in the methods.
  6. Material methods most the citation is the webpage, some website is not working, hence, better to cite the original research paper.
  7. Results must be explained clearly and in detail.
  8. qRT-PCR methodology provided is also very vague and confusing. Please provide more details like what was the calibrator used in the study. I assume the authors have used the control as the calibrator. If so, the authors should not include the control within the bar graph as it represents the fold change between the treated vs control and a fold change of “1” for the ‘control’ doesn’t make any sense.  Also, would be good to provide details on what reagents (details of probes used, if any, if SYBR was used then details for that, etc.) and real time PCR machine were used in the current study.
  9. Comparison of the present results with other similar findings in the literature should be discussed in more detail. This is necessary in order to place this work together with other work in the field and to give more credibility to the present results.  
  10. The authors must add a conclusion section.
  11. References: shall have to correct the whole References according to the ”Instructions for the Authors”, e.g. the Journal name is in italics, the year must be bold and you shall have to use the abbreviated name of the Journals cited.

Round 2

Reviewer 2 Report

Dear Editor,

Thank you for providing the opportunity to review the revised manuscript. The manuscript is improved considerably after revision according to the reviewer's comment. Now this study is a suitable contribution to the IJMS. I recommend the manuscript for publication.

Thank you

With best regards